# Association between gait speed deterioration and EEG abnormalities

**Daysi García-Agustin**[1], **Valia Rodríguez-Rodríguez**[2]*, **Rosa Ma Morgade-Fonte**[3¤], **María A. Bobes**[3], **Lídice Galán-García**[3]

**1** Cuban Centre for Longevity, Ageing and Health Studies, Havana, Cuba, **2** Aston Research Centre for Health in Ageing, Aston University, Birmingham, United Kingdom, **3** Cuban Neuroscience Centre, Havana, Cuba

¤ Current address: Latino Addiction Treatment Centre, Madrid City Council, Madrid, Spain
* v.rodriguez@aston.ac.uk

**Data Availability Statement:** Data is available from the Aston Data Explorer database (accession number: https://doi.org/10.1706/researchdata.aston.ac.uk.00000557).

## Abstract

Physical and cognitive decline at an older age is preceded by changes that accumulate over time until they become clinically evident difficulties. These changes, frequently overlooked by patients and health professionals, may respond better than fully established conditions to strategies designed to prevent disabilities and dependence in later life. The objective of this study was twofold; to provide further support for the need to screen for early functional changes in older adults and to look for an early association between decline in mobility and cognition. A cross-sectional cohort study was conducted on 95 active functionally independent community-dwelling older adults in Havana, Cuba. We measured their gait speed at the usual pace and the cognitive status using the MMSE. A value of 0.8 m/s was used as the cut-off point to decide whether they presented a decline in gait speed. A quantitative analysis of their EEG at rest was also performed to look for an associated subclinical decline in brain function. Results show that 70% of the sample had a gait speed deterioration (i.e., lower than 0.8 m/s), of which 80% also had an abnormal EEG frequency composition for their age. While there was no statistically significant difference in the MMSE score between participants with a gait speed above and below the selected cut-off, individuals with MMSE scores below 25 also had a gait speed<0.8 m/s and an abnormal EEG frequency composition. Our results provide further evidence of early decline in older adults–even if still independent and active—and point to the need for clinical pathways that incorporate screening and early intervention targeted at early deterioration to prolong the years of functional life in older age.

## Introduction

Despite age-related degeneration, older individuals can have a high quality of life if they do not develop cognitive and mobility disorders and become disabled. Yet, disability in older age begins with changes that gradually accumulate and become an incident difficulty [1, 2].

One way to identify early declines is to ask older adults about limitations for performing specific tasks or activities. However, this strategy will depend on their awareness of the

**Funding:** VRR (the corresponding author) received funds from the Global Challenge Research Fund - Aston University internal call (ID-28014), but the funder had no role in study design, data collection and analysis, decision to publish, or preparation of the manuscript.

**Competing interests:** The authors have declared that no competing interests exist.

difficulty, which might be masked by adopting compensatory modifications to accomplish the relevant tasks [1, 2]. Objective assessment of physical performance, on the other hand, provides direct and unbiased information on essential components of physical function and constitutes a reliable, cost-effective method to detect the presence of early changes.

Among the different components of physical function, gait speed (GS) has been found to be predictive of adverse health outcomes and increased mortality [3, 4]. Several studies have also found that GS deterioration is frequently associated with cognitive decline [5, 6] and with transitioning from mild to severe cognitive impairment [7–9]. Based on these known associations, GS has been proposed as a useful screening measure to identify older individuals at risk for developing these types of disorders [10, 11].

However, gait is a process that, besides musculoskeletal and sensorial components, also depends on significant contributions from high-order cerebral areas for planning, execution, and control. Therefore, deterioration of GS in older individuals that cannot be attributed to significant peripheral disorders or uncontrolled systemic disease is likely associated with abnormal brain function as part of its underlying physiopathology. Such a brain dysfunction could also be the antecedent of the cognitive impairment frequently seen in association with GS decline and serve as its preclinical marker.

In this study, we assessed the gait speed and cognitive status of a cohort of functionally independent older adults to look for changes in GS and the possible association with preclinical cognitive impairment. We also recorded the participants' brain electrical activity (electroencephalogram, EEG) at rest to evaluate the integrity of brain function. Given its high sensitivity, EEG is a valuable tool for assessing alterations in cerebral activity associated with functional states and for capturing the dynamic fluctuations linked to cognitive processes, emotions, and neurological conditions [12, 13]. EEG measurements taken during resting states hold predictive value for cognitive task performance [14, 15], as the signal mirrors the functioning of underlying networks.

Our working hypothesis was that a GS decline present in our participants would be associated with a low MMSE score and EEG patterns abnormal for the age. Confirming our hypothesis would support the screening for early declines in functional older people and the inclusion of GS and EEG as part of the assessment pathways aimed at preventing disability at a point where intervention would be most efficient.

## Methods

A cross-sectional observational study was carried out on 95 community-dwelling older adults —over 60 years old—who regularly practised mild exercise in the community with an instructor and did not have evident cognitive impairment. All participants were recruited through their family doctor and provided written informed consent. Those with uncontrolled chronic medical conditions or acute illness were excluded (n = 5). The research project was approved by the Ethics Committee of the Cuban Centre for Longevity, Ageing, and Health Studies.

A gait speed test was performed on all participants using the 4-Meter Walk Test commonly employed in physiotherapy and geriatric clinics. Participants were instructed to stand still behind a starting line marked on the floor and then walk at their usual pace for 6 meters. The first and last meters were used as acceleration and deceleration zones, while the middle four were the testing zone (Fig 1). GS was quantified with a stopwatch as the time spent to cover the testing zone (i.e., 4 meters) and was expressed in meters per second (m/s). The average time of two trials was used in the analysis. Participants were divided into two groups according to their GS value: 'NorGS' was made up of participants who walked 4 m in less than 5 s (i.e., GS > = 0.8 m/s), and 'LowGS' was made up of participants who walked 4 m in more than 5 s (i.e.,

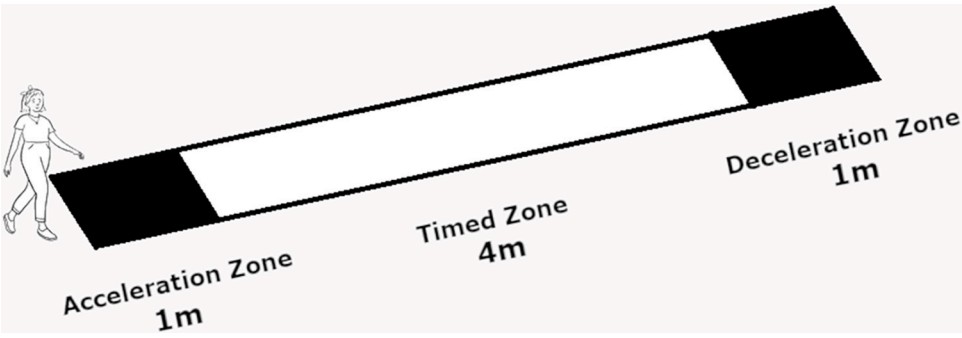

**Fig 1. Diagram representing the gait speed assessment.** Measurements were performed in a flat and unobstructed space by a trained researcher. Participants were instructed to stand at the beginning of the acceleration zone (first black rectangle covering 1 metre) and, upon a command, to walk at their usual pace, passing the deceleration zone (second black rectangle also covering 1 metre). Gait speed was timed with a stopwatch in the middle 4 metres (time zone, the white rectangle in the diagram).

GS < 0.8 m/s). The 0.8 m/s cut-off point was based on recommendations made by several authors [5, 16].

Besides the GS assessment, participants were assessed with the Spanish version of the Mini-mental State Examination (MMSE) and a 10-minute resting-state EEG. The electrophysiological recording was obtained using a 32-channel system (MEDICID 5–32, Neuronic SA) with nineteen Ag/AgCl electrodes placed on the scalp according to the international 10/20 placement system referenced to linked earlobes. Electrode impedance was kept below 10 kΩ. The following acquisition parameters were employed: gain of 10 000, pass-band filters between 0.3–30 Hz, sampling rate of 200 Hz, noise level of 2 μV RMS and environmental temperature of approximately 23˚C. Participants were asked to close and open their eyes at different moments to explore reactivity and avoid drowsiness.

After a standard visual evaluation, a quantitative analysis of the EEG was carried out on the recording of each participant. An expert electroencephalographer (DGA) manually selected artefact-free segments of 2.56 s in duration from eyes-closed periods. About 24 segments obtained from each participant were submitted to a quantitative frequency analysis using a Fast Fourier Transform (FFT). Power spectra were obtained from 0.39 to 19.11 Hz at steps of 0.39 Hz. Absolute Power (AP) in the four classic frequency bands (delta, theta, alpha and beta) —here referred to as broad-band spectral parameters (BBSP)—were calculated and compared to the Cuban normative data [17].

The Z-transformed statistic was used to compare each participant's BBSP against the normative data. This transformation expresses the distance between an individual BBSP and the average BBSP of the normal population for the participant's age. The distance is measured in values of standard deviations (SD), where $|Z| > 1.96$ indicates that the variable is outside the range of the normal population with a 5% risk of error (p<0.05). Therefore, a participant's BBSP was classified as abnormal if the absolute Z value of the AP for any frequency band was above 1.96. A flowchart describing the steps for data processing and analysis is shown in Fig 2.

The possible association between presenting a GS slower than 0.8 m/s and an abnormal quantitative EEG (qEEG) was determined using a Chi-square test. We also used logistic regression to assess whether the GS value predicted the qEEG result. Finally, a frequency-domain source analysis was performed using sLORETA [18]. sLORETA computes the standardised current source density at each of the 6239 voxels in the grey matter and the hippocampus of the MNI-reference brain based on linear weighted sums of the scalp electric potentials. The underlying sources are estimated under the assumption that the neighbouring voxels should

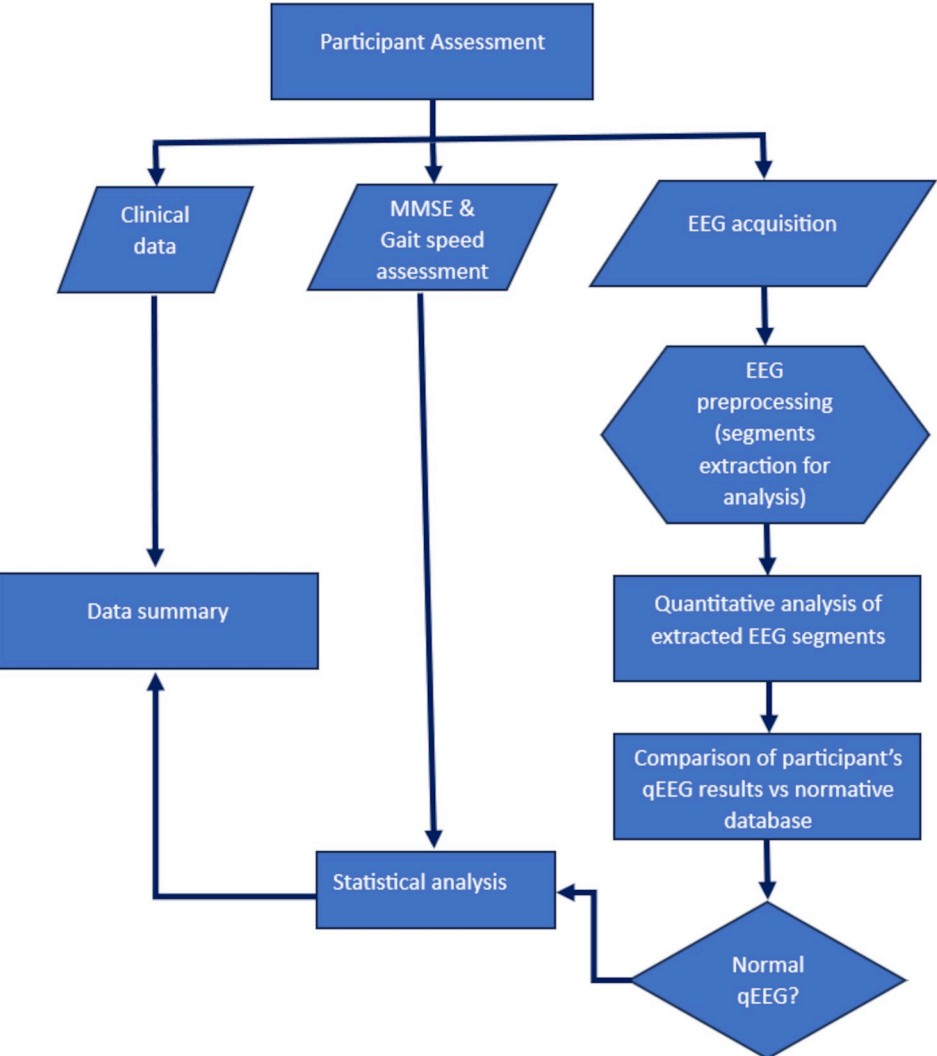

**Fig 2. Flowchart representing the steps followed in this study for the data processing and analysis.**

have maximally similar electrical activity. Differences between the current source densities of LowGS and NorGS in each frequency band are displayed in non-threshold statistical topographical maps using LORETA-KEY software.

## Results

According to the GS value, 27 participants were included in NorGS and 63 in LowGS. Table 1 shows the cohort's characteristics per GS group according to age, sex, known chronic diseases and MMSE score. The distribution of health conditions was similar in both subgroups. Arterial hypertension was present in 41% of the total sample, but other comorbidities (i.e., diabetes and history of stroke) had low prevalence. While some participants scored below 25 in the MMSE in LowGS, 75% of the test scores were at or above this limit (Table 1). There were no significant statistical differences (independent t-test, p>0.05) between NorGS and LowGS in the MMSE.

**Table 1. Characteristics of the cohort included in this study.**

| Cohort Characteristics | NorGS | LowGS |
|---|---|---|
| | **n = 27** | **n = 63** |
| Age | **Mean (SD)** | |
| | 76.8 (8.9) | 80.1 (7.2) |
| MMSE score | **Mean (SD), Min-Max, Percentile 25–75%** | |
| | 29 (1.2), 25–30, 28–30 | 27.8 (2.25), 20–30, 27–30 |
| Female | **Proportion (%)** | |
| | 23 (85.1) | 58 (92.1) |
| Diabetes Mellitus | 3 (11.1) | 10 (15.8) |
| Hypertension | 9 (33.3) | 28 (44.4) |
| History of Stroke | 1 (3.7) | 2 (3.1) |

Fig 3 shows the distribution of participants according to their gait speed and MMSE scores. Note that most of the participants with a GS > = 0.8 m/s and an MMSE score above or equal to 25 had a normal EEG. On the other hand, a larger number of participants with a GS below the 0.8 m/s cut-off or with both lower GS and MMSE below 25 had abnormalities in EEG frequency composition.

Table 2 shows the proportion of normal and abnormal BBSP qEEG studies in both groups. 62 of 90 participants (69%) had EEG frequency abnormalities. Most of these abnormalities (54/62, 87%) were present in participants with GS < 0.8 m/s (LowGS).

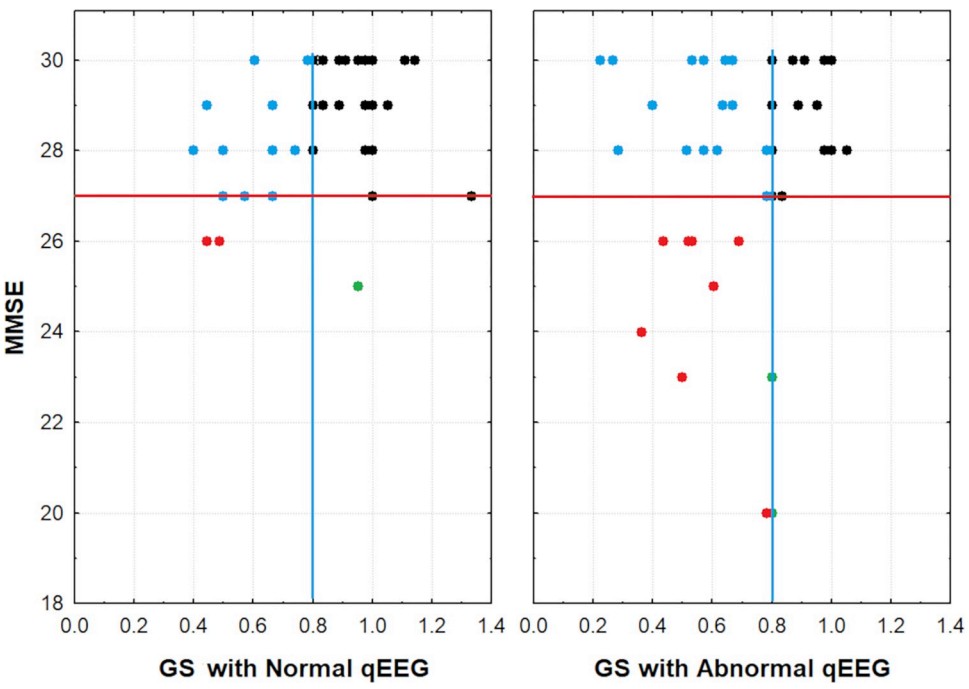

**Fig 3. Distribution of participants according to their gait speed and MMSE scores.** Participants with normal and abnormal EEG frequency composition are represented separately in Panel A and B. The blue line marks the GS cut-off point of 0.8m/s, and the red line marks an MMSE score equal to 25. Black circles: participants with a GS > 0.8 m/s and an MMSE score equal to or above 25. Blue circles: participants with a GS < 0.8 m/s and an MMSE score equal to or above 25. Green circles: participants with a GS > = 0.8 m/s and an MMSE score below 25. Red circles: participants with GS < 0.8 m/s and an MMSE score below 25.

**Table 2. Proportion of normal and abnormal qEEG studies in each gait speed group (qEEG: Quantitative EEG, N: Sample size).**

| Groups | Normal qEEG | Abnormal qEEG |
|---|---|---|
| NorGS (N = 27) | 19 (70.3%) | 8 (29.6%) |
| LowGS (N = 63) | 9 (14%) | 54 (86%) |
| Total (N = 90) | 28 (31.1%) | 62 (68.8%) |

The type of EEG alterations per frequency band in each group is shown in Table 3. An abnormal increment in the energy of delta and theta bands for the age was observed in 37 of 54 individuals (68%) in LowGS. Thirteen participants (24%) in this group also had an abnormal reduction in the energy of the alpha band. On the other hand, while some participants in NorGS (6/27, 22%) had an abnormal increase in the energy of the slow frequency bands, none showed abnormal changes in the alpha power. Otherwise, changes in beta band power were similar in the two groups, with 7% and 6% of participants in NorGS and LowGS, respectively, showing energy increment in this band.

The association between GS < 0.8 m/s and the presence of EEG frequency alterations was statistically significant ($X2(1) = 7.38$, $p < 0.001$; OR 14.25, 95% CI 4.81–42.23) with a positive predictive value of 0.87 and a likelihood ratio of 2.86. The logistic regression analysis also showed that GS values predict the presence of abnormalities in the EEG (Wald (1) = 8.6, $p = 0.003$; OR = 2.96, 95% CI 1.05–5.3).

The distribution of the differences between NorGS's and LowGS's current source maps is shown in Fig 4. As can be seen, the difference in the oscillatory activity was mainly characterised by a widespread increase in theta oscillations in LowGS compared to NorGS, with maxima in the occipitotemporal lobe. LowGS also presented weaker delta oscillations in frontopolar regions. Other oscillatory differences were mild, with LowGS showing increased delta in parietooccipital areas, weaker alpha oscillations in precentral, postcentral and cingulate gyri and a widespread mild increase in beta. Nevertheless, the difference between groups in the oscillatory activity, while present, was not statistically significant.

## Discussion

Our participants comprised older individuals who lived independently and regularly practised mild exercise in community groups with an instructor. Despite that, 70% (63/90) of the total sample had a gait speed slower than 0.8 m/s, of which 86% (54/63) also had abnormal changes in the EEG frequency composition. Furthermore, the gait speed reduction in these individuals was associated with a higher amount of slow frequency activity than expected for their age. Previous research has shown that healthy older individuals have a gait speed above 1 m/s, although this lower limit decreases to ~0.9 m/s in people older than 85 [19]. Values below 0.6 m/s, on the other hand, have been reported in individuals with significant functional and cognitive impairment and are also associated with institutionalisation, hospitalisation, and death

**Table 3. Proportion of abnormal changes per clinical EEG frequency band in each gait speed group.** The frequency band was considered abnormal if its power differed more than 1.96 SD from the population normative data for the corresponding age.

| Groups | ↑Delta | ↑Theta | ↓Alpha | ↑Beta |
|---|---|---|---|---|
| NorGS | 2 | 4 | 0 | 2 |
| LowGS | 15 | 22 | 13 | 4 |
| Total | 17 | 26 | 13 | 6 |

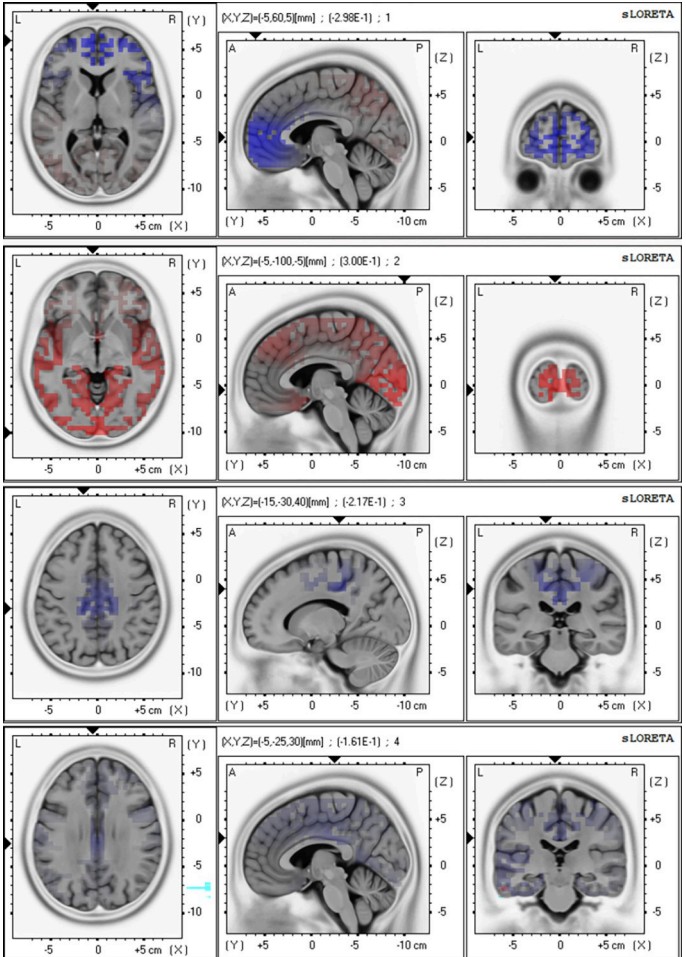

**Fig 4. Topographic maps of the current source difference between groups.** The difference was obtained by subtracting the current source map of NorGS from LowGS (i.e., LowGS—NorGS). Results are plotted by frequency bands on an MNI T2 template of an average brain, with maps centred in the maximum/minimum. Reddish areas are regions where oscillations were stronger in LowGS than in NorGS, while blueish areas are regions with weaker oscillations. First row: delta band: 1–4 Hz, Second row: theta band: 4–8 Hz, Third row: alpha band: 8–12, Fourth row: beta band: 12–30 Hz. L: left, R: right, A: anterior, P: posterior.

[6, 20]. Values lower than the cut-off point used in this study (GS < 0.8 m/s) have been proposed as a risk for developing adverse outcomes [5, 7]. For instance, in a longitudinal study by Montero-Odasso and co-workers (2016), slow gait—defined as walking below 1 m/s at the usual pace—was associated with cognitive impairment and progression to dementia.

In our study, 79% (50/63) of participants with GS < 0.8 m/s (vs 22% (6/27) with GS > = 0.8 m/s) had an abnormally slow EEG. It has been widely demonstrated that EEG background activity slows during physiological ageing. Changes in EEG at an older age are characterised by alpha power reduction and delta/theta power increase [21]. Nonetheless, there is still some debate regarding the extent to which slow activity is a normal pattern in healthy ageing [22]. In this study, EEG frequency changes were considered abnormal only if the individual frequency band deviated more than 1.96 SD from the normative data for the corresponding age.

The association between an early decline in gait speed and an abnormal EEG is not surprising. Gait is a complex function supported by the coordinated interaction of sensorimotor and cognitive processes where several brain areas play a fundamental role [23–25]. A change in the

gait speed of older individuals that are otherwise functionally normal (i.e. no significant musculoskeletal or uncontrolled systemic disorders) must necessarily reflect modifications in the cerebral processes or related networks underlying gait control and implementation [26–28]. On the other hand, as the EEG originates from synchronic neural activity generated across large cortical areas, the signal is sensitive to alterations in brain function. Any disruption in neuronal communication, neural metabolism or changes in the underlying neural network structure will alter the frequency composition of the EEG [29, 30]. Therefore, the presence of abnormal frequency changes in the EEG of our cohort, especially in those participants with slower GS, represents an important subclinical finding as it is a sign of mild brain dysfunction. The presence of brain dysfunction in older individuals with reduced gait speed is also supported by the frequent association between neurological alterations such as delirium and physical deterioration in older persons without primary neurological diseases [31–33].

Several authors have reported increments in the EEG spectral power at slower frequencies (delta and theta) and a decrease in alpha band energy, especially in alpha-1, in individuals with Alzheimer's disease (AD), vascular dementia and mild cognitive impairment (MCI) [21, 34–36]. In particular, higher power in the theta band over the temporal lobe compared to age-matched control has been proposed as a marker of Alzheimer's dementia [37]. Moreover, its association with increased energy in high alpha relative to low alpha frequency has been found to predict the conversion of patients with MCI to AD [38]. In our study, few participants scored below 25 in the MMSE, but those that did also had an abnormal EEG and a GS<0.8 m/s. Nevertheless, as most of our participants with GS deterioration did not have a degree of cognitive decline that the MMSE could identify, we failed to find a relationship between early changes in GS and cognition. Other studies have reported, however, that MMSE has low sensitivity for detecting minor cognitive impairment [39, 40]. Its use represents the main limitation of our study as we cannot rule out that more individuals in our cohort could have had a pre-clinical cognitive decline–especially those with slower gait speed and an abnormal EEG. Nonetheless, another possibility is that our participants were in an early stage where cognitive changes had not yet occurred. It is known from previous studies [41, 42] that GS deterioration can precede the decline in cognition by several years.

We do not know the temporal relationship between the beginning of the gait deterioration and the development of EEG changes. Regardless of the moment when the changes occurred, we found that a GS < 0.8 m/s represented a three times greater risk for having an abnormal EEG and consequently for the presence of brain dysfunction, with a positive predictive value of 87%. Given that GS deterioration and EEG slow frequency composition have been individually identified as predictors of further decline at an older age, their association in the same individual is likely to represent a higher risk of vulnerability to develop conditions such as cognitive frailty or cognitive decline.

In this study we assessed the association between gait speed decline and the presence of brain dysfunction by analysing the EEG at rest. However, we aim to incorporate in future research studies, the simultaneous acquisition of brain electrical activity during walking tasks which was not possible in this investigation. By simultaneously acquiring the EEG and gait, it is possible to elucidate specific changes in brain dynamics related to the different phases of gait. Furthermore, the high temporal resolution of EEG allows us to closely follow the coupling between movement planning, implementation and control and the brain's electrical activity. This strategy is essential to better understand the nature of the gait deterioration and design specific rehabilitation.

Even though participants in our study engaged in regular, mild exercise in the community, the GS decline was present in 70% of the sample. The fact that both groups were mainly composed of women with a mean age of 76.8 (NorGS) and 80 (LowGS) could have been one reason

for the predominance of low GS values. While there was not a significant difference in age between groups, it is known that the level of physical deterioration increases with age. Muscle strength decreases by 1.5% per year and accelerates to 3% per year after 60 years of age [43]. Therefore, the combination of a higher female composition and a mean age of about 80 could have determined lower GS values in the LowGS group. Nevertheless, the prevalence of lower gait values in our study compared to the literature is puzzling. Previous studies conducted in comparable samples have found higher GS values [19, 44–46]. For example, in their 2022 study, Dommershuijsen and colleagues found that individuals aged 89 exhibited GS values higher than 1 m/s for both sexes, while men and women older than 90 showed GS values of 0.9 m/s and 0.8 m/s, respectively. Bohannon and Wang (2019) reported similar findings, with the exception that GS for women aged 80–85 was 0.88 m/s. A study by Lau and co-workers (2020) described normative values higher than 1 m/s in Southeast Asian older adults of both sexes aged below 80 and 0.8 m/s in participants aged 81 and older. Finally, Lusardi's study [46] also found that older women aged 80 and above showed GS values of 0.8 m/s (SD 0.20).

One explanation for the disagreement with the literature could be methodological differences in the GS assessment protocol. While Bohannon and Wang (2019) assessed the gait speed in a 4 m walkway, they did not use an acceleration zone. Different studies have reported that the use of dynamic or static protocols produces different gait results [47–49]. On the other hand, Dommershuijsen, Lau and Lusardi's studies measured the participants' gait with a GaitRite system [19, 45, 46] and 6 m/s walkways [19, 45]. Short (3–4 m) and long (5–10 m) walkway lengths produce reliable results across testing sessions, but they do not have sufficient concurrent validity [50]. The same applies to gait measurements obtained with the GaitRite system and walking tests using a stopwatch. In a study by Peters and co-workers (2015), GS measured with GaitRite showed higher values in all trials performed by community ambulators than in the 3 m walking test [51].

We cannot ignore, however, the socioeconomics of the study sample. Despite being classified as an upper middle-income country according to the World Bank, Cuba's sustained poor economic situation puts individuals in constant hardships. Notwithstanding good health care and social initiatives to improve life, older people are one of the main vulnerable groups affected by pensions of deficient purchasing power, food scarcity, deficient nutrition, deteriorated housing, uncertainty regarding the future and persistent stress. We cannot rule out that these conditions have also influenced our results as it is known that wider determinants, nutrition and well-being impact people's health, including their frailty levels and how they age. A comparative study, which includes cohorts with similar and distinct socioeconomic characteristics to that of our sample, is essential to determine if this is the cause for the observed differences.

## Conclusions

Our results support screening for and intervening in early decline in mobility to prevent or delay the onset of disability and dependence. Specifically, they support the inclusion of gait speed evaluation in the standard assessment of older adults, given its simplicity, sensitivity, and association with underlying brain function. The latter would be beneficial to intervene in gait deterioration early enough to avoid falls, hospitalisation and the evolution towards physical and cognitive decline. Here, we showed that, despite being physically active, older people can present declines in GS (= <0.8 m/s). Slower GS was associated in this study with abnormally slow brain electrical activity for age, representing mild brain dysfunction. The latter poses an additional risk of developing further decline, including cognitive deterioration. Electrophysiological markers such as those used in this study are inexpensive and easy to

implement, which makes it feasible to incorporate them in the functional assessment of older adults contributing to risk stratification. With the challenge of prolonging years of functional life in ageing societies, it is essential to design strategies to restore early declines. A strategy that combines markers such as gait speed and quantitative EEG measures may be especially valuable in objectively assessing the success of intervention programs aimed at rehabilitating changes in mobility.

## Author Contributions

**Conceptualization:** Daysi García-Agustin.

**Data curation:** Valia Rodríguez-Rodríguez.

**Formal analysis:** Daysi García-Agustin, Valia Rodríguez-Rodríguez, Rosa Ma Morgade-Fonte, Lídice Galán-García.

**Funding acquisition:** Valia Rodríguez-Rodríguez.

**Investigation:** Daysi García-Agustin, Valia Rodríguez-Rodríguez.

**Methodology:** Daysi García-Agustin, Valia Rodríguez-Rodríguez, Rosa Ma Morgade-Fonte, María A. Bobes, Lídice Galán-García.

**Project administration:** Daysi García-Agustin.

**Supervision:** María A. Bobes, Lídice Galán-García.

**Visualization:** Valia Rodríguez-Rodríguez.

**Writing – original draft:** Valia Rodríguez-Rodríguez.

**Writing – review & editing:** Daysi García-Agustin, Valia Rodríguez-Rodríguez, Rosa Ma Morgade-Fonte, María A. Bobes, Lídice Galán-García.

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
