## [Decision Letter · Decision Letter 0]

5 Nov 2022

PONE-D-22-18110Hidden decline in older adults: association between gait speed deterioration and EEG abnormalities.

PLOS ONE

Dear Dr. Rodriguez-Rodriguez,

 Thank you for submitting your manuscript to PLOS ONE. After careful consideration, we feel that it has merit but does not fully meet PLOS ONE’s publication criteria as it currently stands. Therefore, we invite you to submit a revised version of the manuscript that addresses the points raised during the review process.

Please note that we have only been able to secure a single reviewer to assess your manuscript. We are issuing a decision on your manuscript at this point to prevent further delays in the evaluation of your manuscript. Please be aware that the editor who handles your revised manuscript might find it necessary to invite additional reviewers to assess this work once the revised manuscript is submitted. 

We look forward to receiving your revised manuscript.

Kind regards,

Alice Coles-Aldridge

Editorial Office

PLOS ONE

Journal Requirements:

Reviewers' comments:

Reviewer's Responses to Questions

**Comments to the Author**

1. Is the manuscript technically sound, and do the data support the conclusions?

Reviewer #1: Partly

2. Has the statistical analysis been performed appropriately and rigorously? 

Reviewer #1: Yes

3. Have the authors made all data underlying the findings in their manuscript fully available?

Reviewer #1: Yes

4. Is the manuscript presented in an intelligible fashion and written in standard English?

Reviewer #1: Yes

5. Review Comments to the Author

Reviewer #1: Overall, the manuscript is written well and in easy to understand manner. The materials are informative and contain some interesting findings. However, some revision is required.

1. Some clarification is required regarding the naming of the groups used in the manuscript.

Lines 158-159: “However, changes in beta band power were similar in the two groups, with 7% and 6% of participants in NorGS and B, respectively, showing energy increment in this band.”

NorGS was defined before and is clear, however, “group B” seems a bit unclear. If it refers to the figure1, panel B, then the samples presented in panel B should be defined somewhere in the text as group B.

Following, in lines 172-173: “The difference was obtained by subtracting the spectral power of NorGS from LowGS (i.e., B-A).”

Meaning that “B” refers to LowGS. Is it the same “B” as before?

2. “Conclusions” section is missing. Some conclusions seem to be drawn in the Discussion section. I recommend separating the two into Discussion and Conclusions sections.

6. PLOS authors have the option to publish the peer review history of their article (what does this mean?). If published, this will include your full peer review and any attached files.

Reviewer #1: No

---

## [Author Response · Author response to Decision Letter 0]

13 Dec 2022

Dear PLOS ONE editorial office:

Please, find below a point-to-point reply to Reviewer#1’s comments.

Sincerely yours,

Valia Rodriguez-Rodriguez

Aston University

Reply to Reviewer #1:

Comment 1. Some clarification is required regarding the naming of the groups used in the manuscript. Lines 158-159: “However, changes in beta band power were similar in the two groups, with 7% and 6% of participants in NorGS and B, respectively, showing energy increment in this band.” NorGS was defined before and is clear, however, “group B” seems a bit unclear. If it refers to the figure1, panel B, then the samples presented in panel B should be defined somewhere in the text as group B.

We used A and B to name the groups in the first draft. For clarity, we renamed A as NorGS (normal gait speed) and B as LowGS (low gait speed). However, we did not replace the ’B’ in Lines 158-159. Thanks to the reviewer for spotting that error that is now corrected.

Comment 2: in lines 172-173: “The difference was obtained by subtracting the spectral power of NorGS from LowGS (i.e., B-A). ”Meaning that “B” refers to LowGS. Is it the same “B” as before?

Yes, ‘B’ refers to ‘LowGS’. As before, we did not replace it. It is corrected and lines 173-174 now read:

‘The difference was obtained by subtracting the spectral power of NorGS from LowGS (i.e., LowGS - NorGS).’

Comment 3: “Conclusions” section is missing. Some conclusions seem to be drawn in the Discussion section. I recommend separating the two into Discussion and Conclusions sections

We did not include Conclusions because the section is optional according to PLOS submission guidelines. Nevertheless, we are now closing the manuscript with a Conclusion paragraph.

---

## [Decision Letter · Decision Letter 1]

28 Jun 2023

PONE-D-22-18110R1Hidden decline in older adults: association between gait speed deterioration and EEG abnormalities.PLOS ONE

Dear Dr. Rodriguez-Rodriguez,

Thank you for submitting your manuscript to PLOS ONE. After careful consideration, we feel that it has merit but does not fully meet PLOS ONE’s publication criteria as it currently stands. Therefore, we invite you to submit a revised version of the manuscript that addresses the points raised during the review process.

Your manuscript has been evaluated by three reviewers: the single reviewer from the previous version, and two new reviewers. Their comments are appended below. The new reviewers have raised a number of concerns, particularly regarding the study design, reporting, and discussion. Please ensure you address each of the reviewers' comments when revising your manuscript.

We look forward to receiving your revised manuscript.

Kind regards,

Hugh Cowley

Staff Editor

PLOS ONE

Reviewers' comments:

Reviewer's Responses to Questions

**Comments to the Author**

1. If the authors have adequately addressed your comments raised in a previous round of review and you feel that this manuscript is now acceptable for publication, you may indicate that here to bypass the “Comments to the Author” section, enter your conflict of interest statement in the “Confidential to Editor” section, and submit your "Accept" recommendation.

Reviewer #1: All comments have been addressed

Reviewer #2: (No Response)

Reviewer #3: All comments have been addressed

2. Is the manuscript technically sound, and do the data support the conclusions?

Reviewer #1: Yes

Reviewer #2: Partly

Reviewer #3: Partly

3. Has the statistical analysis been performed appropriately and rigorously? 

Reviewer #1: Yes

Reviewer #2: Yes

Reviewer #3: Yes

4. Have the authors made all data underlying the findings in their manuscript fully available?

Reviewer #1: (No Response)

Reviewer #2: Yes

Reviewer #3: Yes

5. Is the manuscript presented in an intelligible fashion and written in standard English?

Reviewer #1: Yes

Reviewer #2: Yes

Reviewer #3: Yes

6. Review Comments to the Author

Reviewer #1: The article is well written, has thorough explanation and addresses all important aspects. The authors have addressed all of the prior comments and I have no additional ones. I recommend this article to be accepted.

Reviewer #2: This work explores the gait speed and cognitive status (through MMSE) of 95 older adults. Along with this, a quantitative analysis of their resting state EEG was performed. Results showed that 70% of the sample size had lower gait speed of which 80% have abnormal EEG frequency composition. Overall, the work is fine. However, there are several important points which should be addressed before the final acceptance of the article.

1. The introduction may benefit by highlighting the relevance of using EEG modality.

2. Please cite the previous literature in the introduction which proved that resting state EEG can predict the changes in brain functions while executing the task.

3. Please add a picture of the representative participant while conducting the experiments in the method section. This will improve readability.

4. It would be better if the authors could add a flowchart describing the analysis and data processing of the study.

5. Line 101-102- “Participants were asked to close and open their eyes at different moments to explore reactivity and avoid drowsiness”. I believe that this might have affected the resting state EEG data. Additionally, the authors need to explain why they have not considered the data in eyes open condition. During the trials, the individuals were walking with their eyes open. The authors should consider this.

6. Topographical maps of Beta band are missing in figure 2. Please check.

7. Line 112-113- “This transformation expresses the distance between an individual BBSP and the average BBSP of the normal population for the participant's age.” When did the authors recruit age matched normal population? It is not mentioned in the document anywhere.

8. It will be good if the authors can discuss why they have not measured the participant’s EEG while they were walking at a certain gait speed. Measuring simultaneous EEG and gait speed would be more beneficial. It can help scientists and clinicians to interpret the hidden changes in brain activity more precisely.

9. Along with gait speed, it would be good if the authors could measure other gait parameters such as swing/ stance interval, toe-off and heel strikes, etc. and link these parameters with the changes in the cortical activity. This will add new insights into the study and help the future studies working on the rehabilitation of these individuals.

10. The authors can also consider finding the functional connectivity between highlighted brain regions.

11. The translation of the present findings into clinical applications is not clear. The article would benefit if the authors could explain this point in the conclusion of the study.

Reviewer #3: The study sought to explain whether the “hidden functional changes” in older adults are associated with an early decline in older adults' mobility. Although the study's results were evident, I believe that several issues should be taken into account before drawing a conclusion about the “hidden functional changes” and considering the replies to the authors' hypotheses. I've highlighted a few concerns and suggestions below, particularly for the discussion section:

Line 27: Please consider replacing “doctors” with “allied health professionals”.

Lines 32-33: The information regarding gait speed was not clear in the abstract. “We measured their gait speed at the usual pace (0.8 m/s cut-off point)”. In lines 35-36, the authors stated that 70% of the sample had a preclinical gait speed deterioration (i.e. lower than 0.8 m/s). After all, was 0.8 m/s seen as a sharp cut-off value for the usual pace or as an impaired gait speed? The abstract does not make this clear. Please make this information more understandable. I understand you, firstly, just measured walking speed. After, you consider the value of 0.8 m/s to differentiate the groups based on participants’ gait speed.

Lines 62-63: Please provide at least a reference to this statement: “However, gait is a process that, besides musculoskeletal and sensorial components, also depends on significant contributions from high-order cerebral areas for planning, execution, and control.”

Lines 63-66: Please provide at least a reference to this statement: “A preclinical deterioration of GS in older individuals without significant peripheral disorders or uncontrolled systemic disease is likely to be accompanied by subclinical changes in brain function as part of its underlying physiopathology.”

Methods

Lines 90, and 117: Please standardize the gait unit measure to m/s.

Why did the authors opt for dichotomous data (GS slower than 0.8 m/s: yes or no; normal or abnormal BBSP) and a Chi-square test, rather than continuous data for correlation analyses, for example, by a Pearson or Spearman correlation test?

Results

Table 1 – Please replace “N” with “n”.

Table 1 – Please delete the comma in “Mean (SD),”.

Discussion

Were the study hypotheses confirmed or refuted?

Despite the participants age around 77 years old, living independently in the community, and practicing physical activity with moderate regularity, to what do the authors attribute the percentage of 70% of the sample to have presented walking speed below 0.8 m/s? This is a relatively low gait speed for community-dwelling older adults with a relatively good level of physical activity. Please address this issue in the discussion, based on the literature.

Line 204: The authors stated, “The association between an early decline in gait speed and an abnormal EEG is not surprising.” So what was the additional motivation for this study if this result was already expected? Also, what do the authors consider an “early decline in gait speed”? Older adults with not very advanced ages or individuals with a walking speed that is not very compromised, i.e., above 1 m/s?

Lines 225-227: What do the authors consider “preclinical GS deterioration”? I don't consider a GS below 0.8 m/s as a preclinical characteristic. This is a very considerable and evident gait impairment. Also, “hidden changes in GS”. What do you attribute this term to? GS has been objectively evaluated and has nothing hidden chance. The GS, for my understanding, is very evident. Perhaps hidden changes in electroencephalographic activity and not in GS itself.

Lines 227-229: The authors stated “Different studies have reported, however, that MMSE is not a sensitive tool for detecting minor cognitive impairment [28,29].” Given that, why did the authors choose to use the MMSE instead of MoCA, for example? So, from what I understand, the authors wanted to verify the ability of the MMSE to identify any “preclinical GS deterioration”. If the authors already knew from the literature that the MMSE is not a sensitive tool for detecting minor cognitive impairment, why, even so, did they use this instrument in the study?

Lines 230-231: The authors stated “Another possibility to consider is that our participants were in an early stage where cognitive changes had not yet occurred.” Was it the MMSE test that failed to identify the participants' minor cognitive impairment, or were they actually in this stage of the condition? Perhaps the authors would have had a more accurate tool for identifying mild cognitive impairment if they had employed MoCA, for instance. Due to the potential for a false negative generated by the instrument employed to detect mild cognitive impairment, the authors should take another look at this lack of relationship.

Line 242: “Preclinical GS decline was present in 70% of the sample”. I continue to believe that a gait speed of less than or equal to 0.8 m/s should not be regarded as a preclinical characteristic. According to the research, this is already a significantly reduced walking speed.

Please provide and discuss the study's strengths and limitations.

7. PLOS authors have the option to publish the peer review history of their article (what does this mean?). If published, this will include your full peer review and any attached files.

Reviewer #1: No

Reviewer #2: No

Reviewer #3: No

---

## [Author Response · Author response to Decision Letter 1]

7 Jan 2024

Reply to Reviewer #2:

We are very grateful for your comments and suggestions as they have helped us improve our manuscript. Please find below a point-to-point reply to your comments.

1. The introduction may benefit by highlighting the relevance of using EEG modality. 

2. Please cite the previous literature in the introduction which proved that resting state EEG can predict the changes in brain functions while executing the task.

R/ We added the following text in the introduction:

“Given its high sensitivity, EEG is a valuable tool for assessing alterations in cerebral activity associated with functional states and for capturing the dynamic fluctuations linked to cognitive processes, emotions, and neurological conditions (12,13). EEG measurements taken during resting states hold predictive value for cognitive task performance (14,15), as the signal mirrors the functioning of underlying networks.”

3. Please add a picture of the representative participant while conducting the experiments in the method section. This will improve readability.

R/ Gait speed (GS) assessment was not experimental, but we used a method commonly employed by physiotherapists and geriatricians in the clinical practice. Participants were instructed to stand still behind a starting line marked on the floor (that was the beginning of the acceleration zone; see diagram below) and then walk at their usual pace for 6 meters. The first and last meters were used as acceleration and deceleration zones (represented as black rectangles in the diagram), while the middle four meters constituted the testing zone (represented as a white rectangle in the diagram). GS was quantified by the researcher with a stopwatch as the time spent to cover the testing zone (i.e., 4 meters) and was expressed in meters per second (m/sec).

We added a diagram (Fig. 1), as suggested by the reviewer, with a brief explanation that reads:

Fig 1. Diagram representing the gait speed assessment. Measurements were performed in a flat and unobstructed space by a trained researcher. Participants were instructed to stand at the beginning of the acceleration zone (first black rectangle covering 1 metre) and upon a command, to walk at their usual pace, passing the deceleration zone (second black rectangle also covering 1 metre). Gait speed was timed with a stopwatch in the middle 4 metres (time zone, the white rectangle in the diagram). 

4. It would be better if the authors could add a flowchart describing the analysis and data processing of the study. 

R/ The Fig1S -showing the analysis and data processing flow- is now available as supporting material.

5. Line 101-102- "Participants were asked to close and open their eyes at different moments to explore reactivity and avoid drowsiness". I believe that this might have affected the resting state EEG data. Additionally, the authors need to explain why they have not considered the data in eyes open condition. During the trials, the individuals were walking with their eyes open. The authors should consider this. 

R/ The reviewer correctly assumes that the resting state EEG will differ under eyes closed (EC) and eyes open (EO) conditions. It is known that the frequency composition of the EEG is different under each of these states. Furthermore, Wei et al (2018) found that while brain activity during EC was higher in sensorimotor areas and temporal cortex, there was more activity in posterior areas, including the occipital and parietal cortex during EO. Therefore, the spontaneous fluctuation of brain activity is closely related to EC and EO resting state.

However, EO was only employed in our study as a procedure to guarantee the participant kept awake during the whole recording. Consequently, we did not acquire enough time under EO to perform a frequency analysis. We preferred to acquire the EEG under EC because it is an excellent condition to study brain activity without the influence of visual stimulus driving participants' internal state – as a consequence of internal thoughts triggered by visual analysis and attention. This was especially relevant in our study as we did not have an experimental task that required a baseline at rest with EO.

Another reason behind our decision is that clinical EEG, and most of the literature that have assessed changes in the EEG frequency composition in relation to cognitive status, have used segments during EC. This state enabled us to compare our results with previous findings.

Nevertheless, we will seriously consider the reviewer's concern and include EO as a state of interest in our future studies.

• Wei Jie et at. 2018. Eyes-Open and Eyes-Closed Resting States With Opposite Brain Activity in Sensorimotor and Occipital Regions: Multidimensional Evidences From Machine Learning Perspective. Frontiers in Human Neuroscience, 12. https://doi.org/10.3389/fnhum.2018.00422.

6. Topographical maps of Beta band are missing in figure 2. Please check.

R/ Since the main changes we described in our study comprise delta, theta and alpha bands, we initially presented topographical maps for those frequency bands. We also split the alpha band into lower and higher components to compare against the literature. However, for consistency, we now produced a figure with all frequency bands, including beta and alpha bands from 8-12Hz - as used in our study. We also modified the representation, and instead of surface maps, we are showing results on volumetric maps for better visualisation.

7. Line 112-113- "This transformation expresses the distance between an individual BBSP and the average BBSP of the normal population for the participant's age." When did the authors recruit age matched normal population? It is not mentioned in the document anywhere. 

R/ We did not recruit age-matched controls. Instead, we use the normative database of the Cuban population to perform the comparison according to the participant's age. The database (Valdes-Sosa PA et al, 1990) contains quantitative EEG measures obtained from 211 healthy participants between 5 and 97 years old.

In the Methods section, we specify the following:

'… Absolute Power (AP) in the four classic frequency bands (delta, theta, alpha and beta) - here referred to as broad-band spectral parameters (BBSP) - were calculated and compared to the Cuban normative data of quantitative EEG (qEEG) [13].

The Z-transformed statistic was used to compare each participant's BBSP against the normative data.'

• Valdes-Sosa PA, Biscay R, Galan L, Bosch J, Szava S, Virues T. High resolution spectral EEG norms for topography. Brain Topography. 1990;3:281–2. 

8. It will be good if the authors can discuss why they have not measured the participant's EEG while they were walking at a certain gait speed. Measuring simultaneous EEG and gait speed would be more beneficial. It can help scientists and clinicians to interpret the hidden changes in brain activity more precisely. 

R/ We agree with the reviewer that recording the EEG while the participant is walking is essential to understand the brain dynamics related to the implementation and execution of gait and identify specific electrophysiological changes related to deterioration in waking. Unfortunately, we did not have the necessary technology (eg. portable amplifiers that the participant could carry while walking) to perform this measurement when we conducted this study.

9. Along with gait speed, it would be good if the authors could measure other gait parameters such as swing/ stance interval, toe-off and heel strikes, etc. and link these parameters with the changes in the cortical activity. This will add new insights into the study and help the future studies working on the rehabilitation of these individuals.

R/ We agree with the reviewer that measuring other gait parameters is also essential as they change with age, and their deterioration has been found to be associated with frailty. For instance, Montero-Odasso and co-workers (2011) found that frail older adults, besides a reduced gait speed, presented a high variability in the stride time. However, the assessment of stride time (ms), cadence (steps/min), step width (cm), and double support time (ms) require a technology (e.g. GAITRite) we did not have at the time we performed this study. 

On the contrary, gait speed was a parameter easy to measure in our conditions. The latter it is also one of the points of our study: to support, beyond research, the inclusion of gait speed evaluation in the standard clinical -geriatric- assessment, given its simplicity, sensitivity and association with underlying abnormal brain function.

10. The authors can also consider finding the functional connectivity between highlighted brain regions. 

R/ We sincerely appreciate the reviewer's suggestion. We will conduct a functional connectivity analysis with our data looking at the default mode regions and the sensorimotor regions. However, we wanted to include in this paper only measurements that can be performed in the clinical practice to reach a clinical audience while reserving more 'sophisticated' analysis for another type of paper.

11. The translation of the present findings into clinical applications is not clear. The article would benefit if the authors could explain this point in the conclusion of the study.

R/ We believe our findings can be translated into clinical practice by incorporating walking speed screening in all older adults, even the healthiest. This strategy would be beneficial to intervene in the deterioration of gait early enough and avoid falls or the slow evolution towards physical and cognitive decline. Furthermore, incorporating quantitative EEG into the specialised assessment of mobility function and rehabilitation pathways would allow risk stratification and objective evaluation of intervention success.

We have added a conclusion section were we state the following:

Our results support screening for and intervening in early decline in mobility to prevent or delay the onset of disability and dependence. Here, we showed that, despite being physically active, older people can present declines in GS (=<0.8 m/s). Slower GS was associated in this study with abnormally slow brain electrical activity for age, representing mild brain dysfunction. The latter poses an additional risk of developing further decline, including cognitive deterioration. Electrophysiological markers such as those used in this study are inexpensive and easy to implement, which makes it feasible to incorporate them in the functional assessment of older adults. With the challenge of prolonging years of functional life in ageing societies, it is essential to design strategies to restore early declines. A strategy that combines markers such as gait speed and quantitative EEG measures may be especially valuable in objectively assessing the success of intervention programs aimed at rehabilitating changes in mobility. 

Reply to Reviewer #3 

First, we sincerely appreciate your comments and suggestions as they have helped us improve our manuscript. Please find below a point-to-point reply to your comments.

1- Line 27: Please consider replacing "doctors" with "allied health professionals".

R/ We agree with the reviewer that allied health professionals should be also included as some of them evaluate older adults through different pathways. However, we want to stress that doctors (i.e., physicians such as GPs and Geriatricians) also overlooked early changes as they do not proactively assess older people function to detect subtle modifications. We feel that the term' health professional' includes all categories. Therefore, we changed the sentence, and now it reads: 

These changes, frequently overlooked by patients and health professionals…

2- Lines 32-33: The information regarding gait speed was not clear in the abstract. "We measured their gait speed at the usual pace (0.8 m/s cut-off point)". In lines 35-36, the authors stated that 70% of the sample had a preclinical gait speed deterioration (i.e. lower than 0.8 m/s). After all, was 0.8 m/s seen as a sharp cut-off value for the usual pace or as an impaired gait speed? The abstract does not make this clear. Please make this information more understandable. I understand you, firstly, just measured walking speed. After, you consider the value of 0.8 m/s to differentiate the groups based on participants' gait speed.

R/ We agree with the reviewer that the sentence is not clear. We modified it, and now it reads:

We measured their gait speed at the usual pace and the cognitive status using the MMSE. A value of 0.8 m/s was used as a cut-off point to decide whether they presented a decline in gait speed.

3- Lines 62-63: Please provide at least a reference to this statement: "However, gait is a process that, besides musculoskeletal and sensorial components, also depends on significant contributions from high-order cerebral areas for planning, execution, and control."

R/ The following citations were added:

• Allali G et al (2019). Brain Structure Covariance Associated With Gait Control in Aging. J Gerontol A Biol Sci Med Sci . 74(5):705–13.

• Yogev-Seligmann G, Hausdorff JM, Giladi N. (2008). Movement Disorders. 28:329–42. 

• Holtzer R, Verghese J, Xue X, Lipton RB (2006). Neuropsychology. 20(2):215–23. 

4- Lines 63-66: Please provide at least a reference to this statement: "A preclinical deterioration of GS in older individuals without significant peripheral disorders or uncontrolled systemic disease is likely to be accompanied by subclinical changes in brain function as part of its underlying physiopathology."

R/ The following citations were added:

• Wilson J et al (2019). Neuroscience and Biobehavioral Reviews. Elsevier Ltd; p. 344–69.

• Verlinden VJA et al (2014). Alzheimer's and Dementia. 10(3):328–35. 

• Holtzer R et al (2014). J Gerontol A Biol Sci Med Sci. 69(11):1375–88. 

However, we also modified the sentence, and now it reads: 

"deterioration of GS in older individuals that cannot be attributed to significant peripheral disorders or uncontrolled systemic disease is likely to be associated with abnormal brain function as part of its underlying physiopathology."

5- Methods

Lines 90, and 117: Please standardise the gait unit measure to m/s.

R/ Done

6- Why did the authors opt for dichotomous data (GS slower than 0.8 m/s: yes or no; normal or abnormal BBSP) and a Chi-square test, rather than continuous data for correlation analyses, for example, by a Pearson or Spearman correlation test?

R/ We chose a Chi-square test because while the GS data were continuous, the decision of whether the EEG was normal or not was dichotomous. That is, the participants' quantitative EEG data was classified as abnormal if the absolute Z value of the absolute power for any frequency band was above 1.96; otherwise it was consider as normal. Therefore, we also categorised the GS data using the study cut-off point to assess the association's presence.

Nevertheless, considering the reviewer's suggestion, we have now performed a logistic regression using the GS value as a continuous predictor of abnormality in the EEG. The results showed that GS predicts whether the EEG is abnormal or not (Wald test (1)=8.6, p=0.003; OR= 2.96, CI: 1.05,5.3). This additional test was added to the Method and Results section of the manuscript.

7- Results

Table 1 – Please replace "N" with "n".

R/ Done

8- Table 1 – Please delete the comma in "Mean (SD),".

R/ Done

9- Discussion

Were the study hypotheses confirmed or refuted?

R/ Our hypothesis was that a GS decline in our participants would be associated with a low MMSE score (<25 ) and EEG abnormal for the age. We found that 54 of 63 individuals with GS decline also had a significantly slower EEG. However, ONLY 10 of the 54 older adults with reduced GS and slow EEG had a MMSE < 25. Therefore, our hypothesis was partially confirmed.

10- Despite the participants age around 77 years old, living independently in the community, and practicing physical activity with moderate regularity, to what do the authors attribute the percentage of 70% of the sample to have presented walking speed below 0.8 m/s? This is a

---

## [Decision Letter · Decision Letter 2]

19 Mar 2024

PONE-D-22-18110R2Association between gait speed deterioration and EEG abnormalities.PLOS ONE

Dear Dr. Rodriguez-Rodriguez,

Thank you for submitting your manuscript to PLOS ONE. After careful consideration, we feel that it has merit but does not fully meet PLOS ONE’s publication criteria as it currently stands. Therefore, we invite you to submit a revised version of the manuscript that addresses the points raised during the review process.

Thank you for your effort to improve the manuscript. However, the reviewer still has several minor concerns on this paper. Please consider and respond to the comments from the reviewer.

We look forward to receiving your revised manuscript.

Kind regards,

Ryota Sakurai, Ph.D.

Academic Editor

PLOS ONE

Journal Requirements:

Additional Editor Comments:

Reviewers' comments:

Reviewer's Responses to Questions

**Comments to the Author**

1. If the authors have adequately addressed your comments raised in a previous round of review and you feel that this manuscript is now acceptable for publication, you may indicate that here to bypass the “Comments to the Author” section, enter your conflict of interest statement in the “Confidential to Editor” section, and submit your "Accept" recommendation.

Reviewer #1: All comments have been addressed

Reviewer #2: All comments have been addressed

2. Is the manuscript technically sound, and do the data support the conclusions?

Reviewer #1: Yes

Reviewer #2: Yes

3. Has the statistical analysis been performed appropriately and rigorously? 

Reviewer #1: I Don't Know

Reviewer #2: Yes

4. Have the authors made all data underlying the findings in their manuscript fully available?

Reviewer #1: No

Reviewer #2: Yes

5. Is the manuscript presented in an intelligible fashion and written in standard English?

Reviewer #1: Yes

Reviewer #2: Yes

6. Review Comments to the Author

Reviewer #1: Although authors states that the dataset is available from the Aston Data Explorer, I was unable to locate it. Sharing a link or some guidance on how to access it would be helpful. I don't have any additional comments regarding the manuscript.

Reviewer #2: (No Response)

7. PLOS authors have the option to publish the peer review history of their article (what does this mean?). If published, this will include your full peer review and any attached files.

Reviewer #1: No

Reviewer #2: No

---

## [Author Response · Author response to Decision Letter 2]

5 Apr 2024

Reviewer 1

Reviewer #1: Although authors states that the dataset is available from the Aston Data Explorer, I was unable to locate it. Sharing a link or some guidance on how to access it would be helpful. I don't have any additional comments regarding the manuscript.

Reply: Some information was lacking in the dataset metadata and as a consequence, it was not properly shared. This is now fixed, and the dataset is available here:

https://doi.org/10.17036/researchdata.aston.ac.uk.00000557.

Reviewer 2

1. Flowchart describing the analysis and data processing of the study should be included in the article, not as a supporting information.

Reply: The flowchart is now included as Figure 2. The rest of the figures were re-numbered.

2. Measuring simultaneous EEG and gait parameters can help scientists and clinicians to interpret the hidden changes in brain activity more precisely. Also, the association between the two can be done more accurately when the data is recorded in synchronization. 

I am not really satisfied with the author response on this. I believe the study's protocols has limitations. If the authors can discuss this limitation in detail, it will be good for future research in this field.

Reply: The following paragraph was added acknowledging the benefits of simultaneous acquisition of EEG and gait:

In this study we assessed the association between gait speed decline and the presence of brain dysfunction by analysing the EEG at rest. However, we aim to incorporate in future research studies, the simultaneous acquisition of brain electrical activity during walking tasks which was not possible in this investigation. By simultaneously acquiring the EEG and gait, it is possible to elucidate specific changes in brain dynamics related to the different phases of gait. Furthermore, the high temporal resolution of EEG allows us to closely follow the coupling between movement planning, implementation and control and the brain's electrical activity. This strategy is essential to better understand the nature of the gait deterioration and design specific rehabilitation.

---

## [Decision Letter · Decision Letter 3]

12 Apr 2024

Association between gait speed deterioration and EEG abnormalities.

PONE-D-22-18110R3

Dear Dr. Rodriguez-Rodriguez,

We’re pleased to inform you that your manuscript has been judged scientifically suitable for publication and will be formally accepted for publication once it meets all outstanding technical requirements.

Kind regards,

Ryota Sakurai, Ph.D.

Academic Editor

PLOS ONE

Additional Editor Comments (optional):

Reviewers' comments:

Reviewer's Responses to Questions

**Comments to the Author**

1. If the authors have adequately addressed your comments raised in a previous round of review and you feel that this manuscript is now acceptable for publication, you may indicate that here to bypass the “Comments to the Author” section, enter your conflict of interest statement in the “Confidential to Editor” section, and submit your "Accept" recommendation.

Reviewer #2: All comments have been addressed

2. Is the manuscript technically sound, and do the data support the conclusions?

Reviewer #2: Yes

3. Has the statistical analysis been performed appropriately and rigorously? 

Reviewer #2: Yes

4. Have the authors made all data underlying the findings in their manuscript fully available?

Reviewer #2: Yes

5. Is the manuscript presented in an intelligible fashion and written in standard English?

Reviewer #2: Yes

6. Review Comments to the Author

Reviewer #2: All the comments have been addressed. I accept the manuscript in its current form. I wish the authors all the best for their future endeavors.

7. PLOS authors have the option to publish the peer review history of their article (what does this mean?). If published, this will include your full peer review and any attached files.

Reviewer #2: No

---

## [Editor Report · Acceptance letter]

24 May 2024

PONE-D-22-18110R3 

PLOS ONE

Dear Dr. Rodriguez-Rodriguez, 

I'm pleased to inform you that your manuscript has been deemed suitable for publication in PLOS ONE. Congratulations! Your manuscript is now being handed over to our production team.

Kind regards, 

on behalf of

Dr. Ryota Sakurai 

Academic Editor

PLOS ONE